# Experimental Study of Leakage Monitoring of Diaphragm Walls Based on Distributed Optical Fiber Temperature Measurement Technology

**DOI:** 10.3390/s19102269

**Published:** 2019-05-16

**Authors:** Tao Liu, Wenjing Sun, Hailei Kou, Zhongnian Yang, Qingsheng Meng, Yuqian Zheng, Haotong Wang, Xiaotong Yang

**Affiliations:** 1Shandong Provincial Key Laboratory of Marine Environment and Geological Engineering, Ocean University of China, Songling Road No. 238, Qingdao 266100, China; ltmilan@ouc.edu.cn (T.L.); sunwenjing@stu.ouc.edu.cn (W.S.); hlkou@ouc.edu.cn (H.K.); zhengyuqian@stu.ouc.edu.cn (Y.Z.); whtluck@163.com (H.W.); yxt937431299@163.com (X.Y.); 2Pilot National Laboratory for Marine Science and Technology, Wenhai Road No.1, Qingdao 266003, China; 3School of Civil Engineering, Qingdao University of Technology, Fushun Road No.11, Qingdao 266033, China; yzn1101@163.com

**Keywords:** distributed optical fiber temperature measurement technology, diaphragm wall, leakage monitoring, model test

## Abstract

In geotechnical engineering seepage of diaphragm walls is an important issue which may cause engineering disasters. It is therefore of great significance to develop reliable monitoring technology to monitor the leakage. The purpose of this study is to explore the application of a distributed optical fiber temperature measurement system in leakage monitoring of underground diaphragm walls using 1 g model tests. The principles of seepage monitoring based on distributed optical fiber temperature measurement technology are introduced. Fiber with heating cable was laid along the wall to control seepage flow at different speeds. The temperature rise of the fiber during seepage was also recorded under different heating power conditions. In particular the effect of single variables (seepage velocity and heating power) on the temperature rise of optical fibers was discussed. Test results indicated that the temperature difference between the seepage and non-seepage parts of diaphragm wall can be monitored well using fiber-optic external heating cable. Higher heating power also can improve the resolution of fiber-optic seepage. The seepage velocity had a linear relationship with the final stable temperature after heating, and the linear correlation coefficient increases with the increase of heating power. The stable temperature decreased with the increase of flow velocity. The findings provide a basis for quantitative measurement and precise location of seepage velocity of diaphragm walls.

## 1. Introduction

Urban construction projects such as high-rise building projects have promoted the development and utilization of underground spaces. This also makes the foundation pits development a larger and more important matter [1]. Underground diaphragm walls are a common retaining structure used in foundation pits located in soft soil areas. Because of their complex structure and difficult construction, leakage problems often occur. These problems often cause serious hidden dangers to the safety of geotechnical engineering projects [2]. Therefore, it is of great significance to monitor the seepage of diaphragm wall during construction. Conventional monitoring methods, such as construction borehole inspection, light dynamic penetration, surrounding well water pressure and pumping tests, etc., can only reflect the local conditions of a diaphragm wall and cannot carry out continuous monitoring, and have many limitations during the actual construction process. Many techniques for monitoring seepage in geotechnical engineering have been proposed. Typical methods include: electromagnetic methods [3,4], thermal impedance methods [5] and resistivity methods [6,7,8]. These monitoring methods each have their own advantages and scope of application. However, the above monitoring methods cannot meet the requirements of diaphragm wall monitoring, to locate seepage points and measure seepage velocity, so they are not suitable for diaphragm wall monitoring.

In recent years, distributed optical fiber sensing technology has been widely used in engineering monitoring because of its flame-proof, explosion-proof, corrosion-resistant, electromagnetic interference-resistant, high voltage-resistant, long-distance measurement, real-time measurement and positioning features [9]. Some studies applying distributed optical fiber sensing technology to monitor the leakage in geotechnical engineering projects were also conducted. Zhu et al. [10] proposed a distributed temperature sensing (DTS) technology based on Brillouin scattering light to monitor dam leakage. Tylerd et al. [11] put forward the singular value decomposition method in the process of DTS temperature measurement. Khan et al. [12] designed an automatic monitoring system for abnormal seepage points in a dam based on DTS technology. Because the temperature can be transmitted through the medium and the change of the medium is continuous, the seepage field can be understood by the changes of the medium temperature [13,14,15]. Minardo et al. [16] reported distributed temperature measurements in a perfluorinated graded-index polymer optical fiber (POF) with 50-µm core diameter for the first time. They showed that Brillouin optical frequency-domain analysis was able to resolve spatially the temperature-dependent Brillouin frequency shift profile along a 20-m POF fiber sample, at a nominal spatial resolution of 4 m. Saxena et al. [17] presented a Raman optical fiber distributed temperature sensor, using a wavelet transform-based signal processing technique for backscattered anti-Stokes and Stokes signals. The proposed technique enables automatic measurement of distributed temperature profile that has better temperature accuracy and very small spatial errors in detecting the location of hot zones. However, distributed optical fibers are insensitive to small temperature variations and are susceptible to environmental temperature in the case of small seepage. This defect limits their application in leakage monitoring of geotechnical engineering projects. At present, distributed optical fiber temperature measurement technology can only be applied to qualitative monitoring of seepage fields with large temperature differences and flow velocity, and cannot meet all the requirements of seepage field monitoring.

In this paper, a new method based on distributed optical fiber temperature measurement technology was proposed to monitor the leakage of diaphragm walls. Model tests under different seepage flow velocity and heating power conditions were carried out. The effects of heating power and seepage velocity were discussed. On the basis of test results, a qualitative relationship between seepage velocity and temperature rise is deduced. The test results can provide a theoretical basis and method for distributed optical fiber monitoring of underground diaphragm wall seepage.

## 2. Distributed Optical Fiber Sensing Technology

Distributed optical fiber temperature sensors (DOTSs) can measure the temperature based on the change of optical fiber length in the form of a continuous function of distance [18]. The DOTS principle is based on the Raman scattering of light inside the optical fiber. Using optical time domain reflection technology, a high-power narrow-band optical pulse is sent into the optical fiber. The backscattered light intensity changes with time can then be detected. Rayleigh scattering is the main factor causing the attenuation of optical fiber transmission. Although the backscattering effect is strong, it does not change significantly at the temperature of conventional optical fibers. Raman scattering and Brillouin scattering are much weaker than Rayleigh scattering in intensity, but they are directly related to temperature [19,20,21]. The different spectral distributions are shown in Figure 1.

In Raman scattering, the temperature sensitivity of the anti-Stokes scattering signal is higher than that of Stokes scattering signal. In practical applications, the ratio of anti-Stokes signal to Stokes signal is often used as temperature information to reduce the influence of light source intensity, light injection conditions, geometrical size and structure of optical fibers. The ratio of anti-Stokes to Stokes intensity *R*(*T*) can be calculated by Equation (1): (1)$$R\left(T\right)={\left(\frac{\gamma_{s}}{\gamma_{\mathrm{as}}}\right)}^{4}e^{-\frac{hc\Delta \gamma }{kT}}$$
where $\gamma_{s}$ is the anti-Stokes frequency; ${\gamma_{a}}_{s}$ is the Stokes frequency; *c* is the speed of light in vacuum; Δγ is Raman frequency shift; *h* is Planck constant; *k* is Boltzmann constant, and *T* is the absolute temperature of the environment. It can be seen from Equation (1) that the ratio of anti-Stokes to Stokes intensity *R*(*T*) in Raman scattering is only a function of temperature. This is the fundamental theoretical basis of distributed optical fiber temperature sensor.

## 3. Test Set-Up

### 3.1. Seepage Monitoring Based on DTS

Diaphragm walls are the retaining structures of deep foundation pits, which bear large lateral water and soil pressures. During the grouting process, it is difficult to achieve complete filling due to the influence of embedded parts in the steel cage and the grouting materials [22,23]. Leakage can easily occur at positions where the filling is not dense or two walls connect. In the case of low seepage velocity, the temperature difference between the seepage location and surrounding medium is small. It is difficult to measure this kind of temperature difference using a distributed temperature measurement system. Hence, as shown in Figure 2, in this study the temperature field was superimposed on the temperature measurement optical fiber with a heating cable to increase the signal-to-noise ratio of the DTS system. The parameters of the heating cables used are listed in Table 1.

As the leakage site has a large heat capacity, the temperature change rate is slower for the same heating time. This will lead to a large difference between the fiber temperature around the leakage site and that of a non-leakage site. The temperature change is also closely related to the seepage velocity. At the same heating power, the faster the seepage velocity is, the lower the temperature rise is. According to the measured distance of the DTS, the time from signal transmission to signal reception is determined. Then the light speed in the fiber is determined and the distance is calculated. The distance *X* from any point on the optical fiber can be calculated by Equation (2). The leakage of diaphragm walls can be effectively monitored and located by the temperature change determined by DTS.
(2)$$X=\frac{cT^{\prime }}{2n}$$
where *c* is the light speed in vacuum; *T’* is the time from signal transmission to signal reception, and *n* is the refractive index of the measured light.

### 3.2. Model Test Scheme

In this study, a model box was used for the test. As shown in Figure 3, the model box is 1.0 m × 0.6 m × 0.6 m in volume. It can be seen in Figure 4 that a concrete wall made of C30 concrete was used as the diaphragm wall in the model test. Each cubic meter of C30 concrete is composed of 175 kg of water, 461 kg of cement, 512 kg of sand and 1252 kg of stone. The mix ratio of these materials was 0.38:1:1.11:2.72. The thickness of the concrete wall was 5 cm. The diameter of fiber core was 3 mm and it was combined with a heating cable. The heating cable was uniformly arranged on the concrete wall in an “S” shape. Each section of fiber was numbered 1–6. It can be seen in Figure 5 that four leakage holes with a diameter of 5 cm were prefabricated on the concrete wall. The leakage was located on the left and right sides of the No. 3 optical fiber. Detailed information of the leakage is shown in Figure 4. In this model test, saturated silty clay from a soft foundation pit in the Jinan area was used as filling material to simulate actual geological conditions. It can be seen in Figure 6 that the water tank was connected to the leakage of diaphragm wall through a PVC pipe. The angle between the PVC pipe and the concrete wall was 30°. The PVC pipe was 50 cm long and 10 cm in diameter. The heating system was composed of a heating cable, an alternating current power supply with a voltage regulator used to regulate the voltage and a multi-functional voltmeter used to display the voltage to control the heating power. During the test, heating cables were tightly bundled with the temperature optical fiber. After electrification, the temperature of optical fiber was higher than that of the surrounding environment in order to monitor leakage. A TDGC2-5KVA single-phase voltage regulator manufactured by Zhejiang Fujian Electrical Appliances Co., Ltd. (Wenzhou, China) was used in this test. Its basic parameters are shown in Table 2.

A box-type distributed optical fiber temperature measurement system produced by Suzhou Nanzhi Sensing Technology Co., Ltd. (Suzhou, China) was used as the data acquisition and transmission system. It was selected for its high measurement accuracy and validity. It has characteristics of high measurement accuracy, short measurement time and long measurement distance. The basic parameters of the used DTS system are shown in Table 3.

During the test, the model box was filled with silty clay to simulate the real stratum. The water valve was opened to adjust the seepage rate and the heating power of used cable was adjusted many times, and the temperature of optical fiber was monitored. Specific test schemes are shown in Figure 3, Figure 4, Figure 5 and Figure 6.

### 3.3. Test Process

In the preliminary preparation stage, the larger particles in the model box were removed, the silty clay surface was smoothed, the concrete wall was placed, and the same silty clay as the model box was added to the PVC pipe. According to the optical fiber burial method described above, the optical fiber was laid in an "S" shape. In the course of laying the fiber, it is necessary to ensure that the temperature measuring optical fibers are tightly bundled with the heating optical cables. The optical fibers were then tested. Then we connect the optical fiber with the DTS and set the coordinates. The optical fibers were fixed. After the temperature measuring optical fibers, heating cables and leakage points were basically in the same horizontal plane, soil was slowly added to the model box to the designated level. During the process of soil addition, attention should be paid to the protection of the optical fibers. 

The heating power of cable was set to 4, 6, 8, 10 and 12 W/m. According to Ohm’s law:(3)$$P=\frac{U^{2}}{R}$$

The required heating voltage, set power and corresponding heating voltage were calculated as shown in Table 4.

The voltage regulator was adjusted until the voltmeter showed the required voltage. Then the flow rate control valve was opened to drain water at the specified seepage rate. When the temperature rise of the optical fiber was less than 0.5 °C in 10 min, the heating power was turned off. The temperature of the optical fibers gradually returned to its initial state. In this process, the DTS was used to continuously record the temperature of the optical fibers in different periods. Thereafter, the heating power was adjusted to different values and the above steps were repeated. Different seepage velocities were controlled by adjusting the flow rate control valve and calculating the flow rate through the seepage flow. The flow velocities from *v* = 50 mm/h to *v* = 250 mm/h were designed respectively. The interval of seepage velocities was 50 mm/h. Under the specified heating power, the DTS was used to continuously record the temperature of optical fibers in different time periods. Then different heating powers were set up and the above steps were repeated. Finally, the data were analyzed and processed.

## 4. Results and Discussions

### 4.1. Temperature Change During Monitoring

The overall temperature rise of the optical fiber temperature measurement system under different seepage velocities is shown in Figure 7, Figure 8, Figure 9, Figure 10, Figure 11 and Figure 12. As mentioned above, the leakage point in this test is located at the top of No.3 optical fiber, about 1.7–2 m away from the left end of the optical fiber. Figure 7 shows the temperature change of the optical fiber temperature measurement system when the seepage velocity is 0. It suggests that when there is no leakage in the concrete wall, the temperature of each section of the optical fiber is basically the same. There is no obvious temperature anomaly. This indicates that the heating cable used in this test can uniformly heat all parts of the optical fiber. Figure 8 shows the situation when the seepage velocity is 50 mm/h. It suggests that when the heating power is less than 8 W, the seepage point cannot be clearly distinguished at this seepage velocity. When the heating power is 10 W and 12 W, the temperature rise at the leakage site is 0.5–1 °C lower than that at the adjacent section. Figure 9 shows the situation when the seepage velocity is 100 mm/h. It suggests that when the heating power is greater than 6 W, the temperature rise at the leakage site is 1–1.5 °C lower than that at the adjacent section. The detection effect of the leakage point is better. However, when the heating power is 4 W, the location of leakage point cannot be clearly distinguished. Figure 10 shows the situation when the seepage velocity is 150 mm/h. When the heating power is greater than 6 W, the abnormal temperature rise at the leakage point is more obvious. Figure 11 and Figure 12 show that the abnormal temperature rise becomes more obvious with the increase of heating power. However, when the heating power is 4 W, the abnormal temperature rise will not become more obvious with the increase of seepage velocity. It can be seen that if there is no leakage point and water content in the saturated clay medium is stable, the temperature rise curve of optical fibers can be maintained in a relatively stable state. First, it fluctuates in a small range above and below a certain temperature. After leakage, water participates in the heat transfer process between temperature measuring optical fibers and porous media. Because of the large specific heat capacity of water, the temperature change rate of the leakage part is slower under the same heating time, which results in the difference between the temperature of temperature measuring optical fibers at the leakage site and the temperature without leakage, and the temperature rise at the leakage site will be lower than that at the non-leakage site.

Interestingly, the most significant abnormal temperature rise point in Figure 9, Figure 10 and Figure 11 was observed not at the position of No. 3 optical fiber, but closer to the left end of the optical fiber, and the offset distance is about 0.4 m. This result is somewhat counterintuitive. After the inspection of the test device, it was found that the bottom of the model box was not truly horizontal due to the processing technology used, and the right side was about 0.5 cm higher than the left side. This resulted in a slow flow of water to the left end of the optical fiber at the leakage hole. Therefore, in the following analysis, the lowest temperature change point within 0.4 m from No. 3 optical fiber was selected as the actual leakage point.

The temperature rises under different seepage flow velocities are shown in Figure 13, Figure 14 and Figure 15. Figure 13 is the temperature rise curve at different seepage flow velocity when the heating power is 4 W. It suggests that the optical fiber has poor resolution for the temperature change at the leakage point under a low heating power. The leakage point detection effect is poor. Figure 14 and Figure 15 show that when the heating power is gradually increased to 12 W, the resolution of optical fibers to seepage velocities is obviously improved. The detection effect becomes better with the heating power becomes higher. The conclusion is that the slight change of flow velocity will not cause the obvious change of temperature rise curve. When the seepage flow velocity changes significantly, the corresponding seepage points can be found through the temperature rise curve. Figure 13, Figure 14 and Figure 15 suggest that when the heating power is 4 W, the temperature rise of optical fibers in the area without leakage is also uneven. This reflects that the spontaneous Raman scattering of the optical fiber is less sensitive to the change of lower temperature. Therefore, in the process of engineering application monitoring, high heating power should be selected as far as possible within a reasonable range.

### 4.2. Relationship Between Temperature and Flow Velocity 

The water flow at the leakage point continuously takes away the heat around the optical fiber, a process that is closely related to the seepage velocity. When the heat generated by the heating cable equals the heat taken away by the seepage, the optical fiber will reach a final stable temperature. The seepage velocity under 12 W heating power is compared to analyze the relationship between the temperature change of optical fiber and the seepage velocity. Its relationship graph is shown in Figure 16.

It can be seen from Figure 16 that when the seepage velocity is small, the temperature of the optical fiber in the seepage position rises faster, and the final stable temperature is higher, but it takes more time to reach the final stable temperature, and when the heating stops, the time required for the temperature to decrease is also longer. When the seepage velocity is high, the temperature of the fiber in the seepage position rises slower, and the final stable temperature is lower, but it takes less time to reach the final stable temperature. According to the law of thermodynamics, when the system reaches its final stable temperature, the following heat equation should be satisfied: (4)$$\left\{\begin{array}{l} Q_{1}=\frac{U^{2}}{R}\\ Q_{2}=S_{w}\alpha_{w}\left(T_{end}-T_{0}\right)\Delta l\\ Q_{3}=S_{s}\alpha_{s}\left(T_{end}-T_{0}\right)\\ Q_{1}=Q_{2}+Q_{3} \end{array}\right\}$$
where *Q_1_* is the heat generated by the heating cable; *Q_2_* is the heat exchanged between the optical fiber system and the seepage; *Q_3_* is the heat exchanged between the optical fiber system and the silty clay; *S_w_* and *S_s_* are respectively the heat exchange area of water and soil with the optical fiber system, *S_w_* = *nS_0_*, *S_s_* = (1 −* n)S_0_*, *n* is the porosity of silty clay, *S_0_* is the surface area of the optical fiber system; *α_w_* and *α_s_* are respectively the heat exchange coefficients of water and soil with the optical fiber system; *T_end_* is the final stable temperature; *T_0_* is the initial temperature; Δ*l* is the length of water flowing through the optical fiber system in a unit time.

According to Xiao’s et al. [24] research, the formula for calculating the heat transfer coefficient of fluid and heating system is shown in Equation (5): (5)$$\alpha_{w}=Du$$
where *D* is the characteristic number of the process, it is related to the thermal conductivity of fluids, thermal conductivity of optical fibers, Reynolds number of the seepage and the diameter of optical fibers; *u* is the velocity of the seepage. 

By sorting out the above formulas, Equations (6) and (7) can be obtained:(6)$$T_{end}-T_{0}=\frac{U^{2}}{RS_{0}\left[nDu\Delta l+\left(1-n\right)\alpha_{s}\right]}$$
(7)$$u=\frac{U^{2}}{\left(T_{end}-T_{0}\right)nD\Delta lRS_{0}}+\frac{\left(1-n\right)\alpha_{s}}{nD\Delta l}$$

According to Equations (6) and (7), the seepage velocity can be quantitatively measured after a series of parameters such as heating voltage, heating process temperature rise and heating resistance are measured. The time of reaching the final stable temperature under different seepage velocities is fitted. As shown in Equation (8): *y* = 0.204*x* + 1.511(8)

According to Equation (8), the seepage velocity can be estimated based on the time when the optical fiber reaches the final stable temperature. In the leakage monitoring of diaphragm wall in practical engineering, the optical fiber system can be calibrated beforehand, and then prefabricated on the diaphragm wall. The seepage velocity can be calculated by measuring the final stable temperature and the time to reach the temperature. In order to further study the relationship between temperature rise of optical fibers and seepage velocity, the relationship diagram of the two under different heating power was made. As shown in Figure 17, the temperature rise of the optical fibers near the seepage point is taken as the ordinate and the velocity of the flow as the abscissa. 

According to Yan’s et al. [25] research, the seepage velocity is inversely proportional to the temperature change of the point in the geotechnical seepage field. The rise of the temperature increases with the decrease of the seepage velocity. The curves of the seepage velocity and the temperature rise are fitted under the heating power of 12, 10 and 8 W. When the heating power is 12 W, the correlation coefficient is 0.9536, and the slope of the curve is larger. This indicates that the response of the optical fiber system to the seepage velocity is more sensitive at a higher heating power. When the heating power is 10 W, the coefficient is 0.8658. When the heating power is 8 W, the coefficient is 0.5353. The correlation coefficient of fitting curve increases with the increase of heating power. Therefore, if the seepage velocity is calculated according to the fitting curve, the larger heating power should be selected as far as possible.

## 5. Conclusions

(1) Our physical model tests prove that the distributed optical fiber temperature measurement technology with external cable can accurately locate the leakage point and distinguish between different seepage velocities in underground diaphragm wall seepage monitoring. This technology has a good monitoring effect. 

(2) During the test, the temperature of optical fibers continues to rise due to the existence of heating cables. The temperature of the leaking part changes slowly. This results in a large difference between the temperature of the leaking part and the non-leaking part, and a low temperature rise of the leaking part. When the seepage velocity is low, the temperature rise of each section of the optical fiber is basically the same, and the seepage position cannot be distinguished basically. When the seepage velocity reaches 150 mm/h, the temperature rise of the optical fiber at the seepage point is obviously lower than that at the non-seepage point.

(3) When the heating power of the external cable is small, the resolution of the monitoring system to the seepage velocity is poor, so as high a heating power as possible should be chosen within a reasonable range to improve the monitoring effect.

(4) After heating, the temperature of the optical fiber increases continuously and eventually reaches a stable temperature. In this state, the heat generated by the optical fiber system is equal to the sum of the heat absorbed by the leakage and the heat absorbed by the surrounding soil. When the seepage velocity is high, the stable temperature is lower and it takes less time to reach the stable temperature. When the seepage velocity is low, the stable temperature is higher and the time to reach the temperature is longer. The relationship between seepage velocity and parameters related to final stable temperature, optical fiber and surrounding soil is established. It is hopeful that the quantitative measurement of seepage velocity can be realized. There is a linear relationship between seepage velocity and final stable temperature after heating. The correlation coefficient increases with the increase of heating power. 

(5) It should be noted that the seepage velocity in this experiment is the average velocity over a period of time, and the temperature rise of the optical fiber is the average value of the temperature rise of the whole section of optical fiber. The relationship between the temperature rise of optical fiber and the seepage velocity can only be qualitatively analyzed. For further analysis, we need to use instantaneous velocity meter and more precise temperature measurement system to establish the mathematical relationship between temperature rise of optical fiber and seepage velocity.

## Figures and Tables

**Figure 1 sensors-19-02269-f001:**
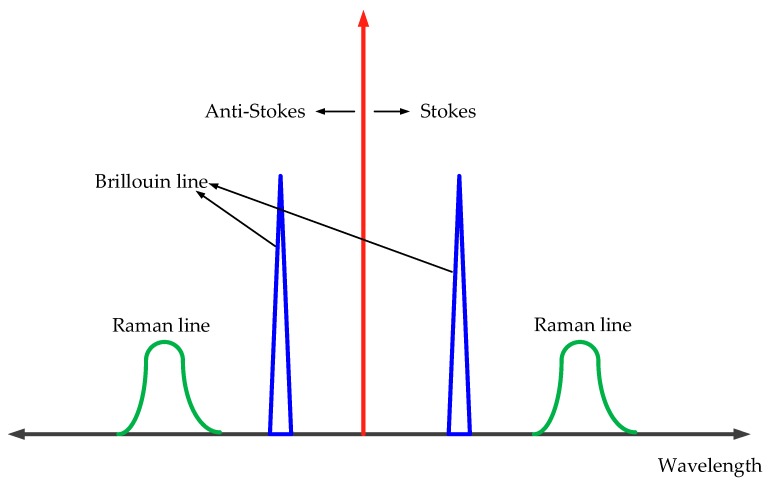
Scattering spectrogram in optical fiber.

**Figure 2 sensors-19-02269-f002:**
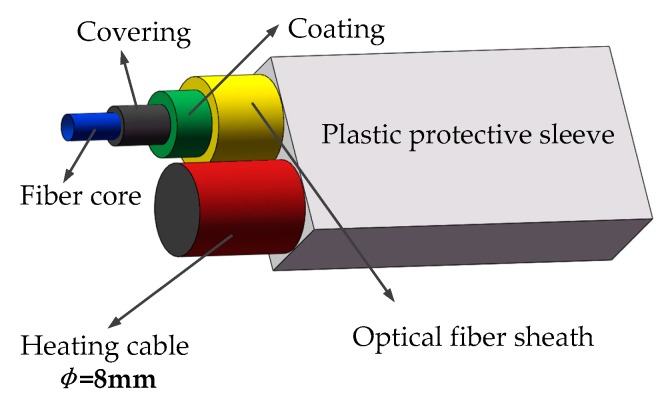
Heating cable and fiber diagram.

**Figure 3 sensors-19-02269-f003:**
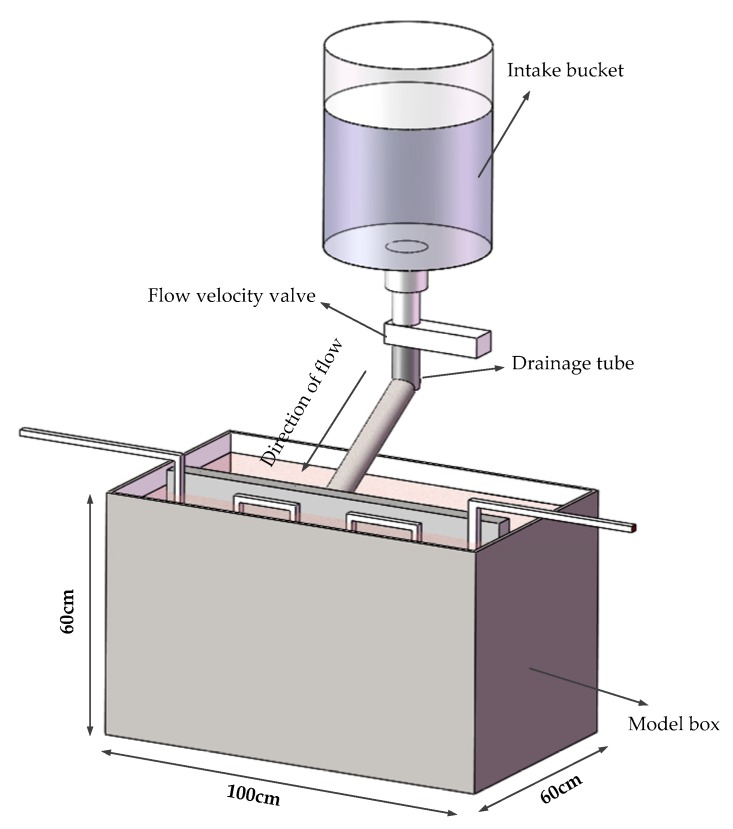
Heating cable and fiber diagram.

**Figure 4 sensors-19-02269-f004:**
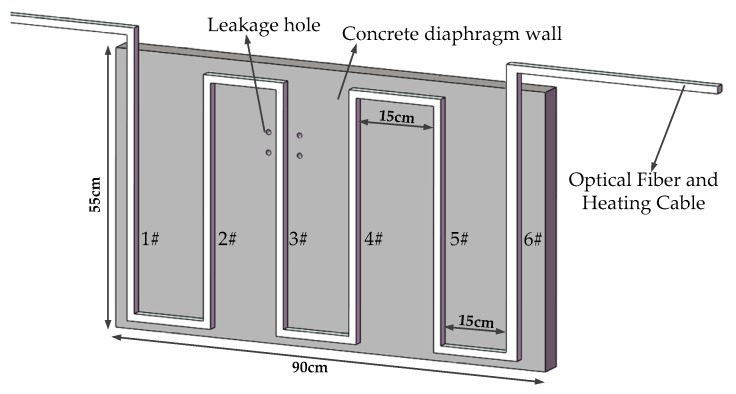
Optical fiber layout.

**Figure 5 sensors-19-02269-f005:**
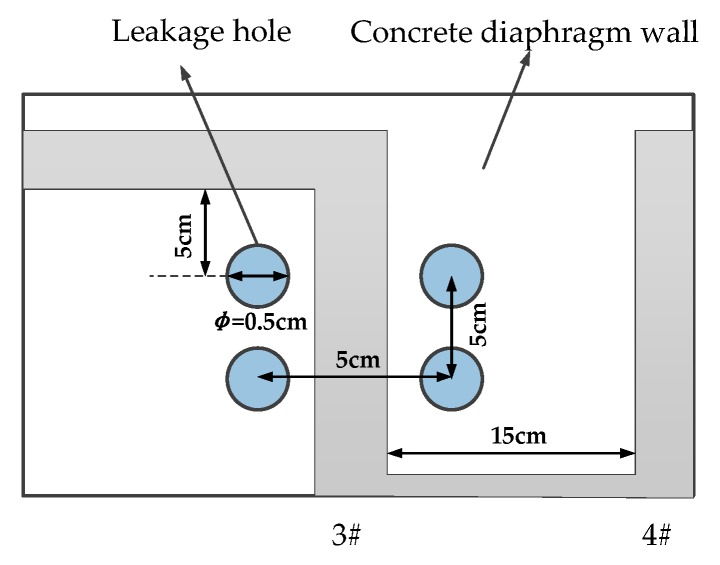
Layout of leakage holes.

**Figure 6 sensors-19-02269-f006:**
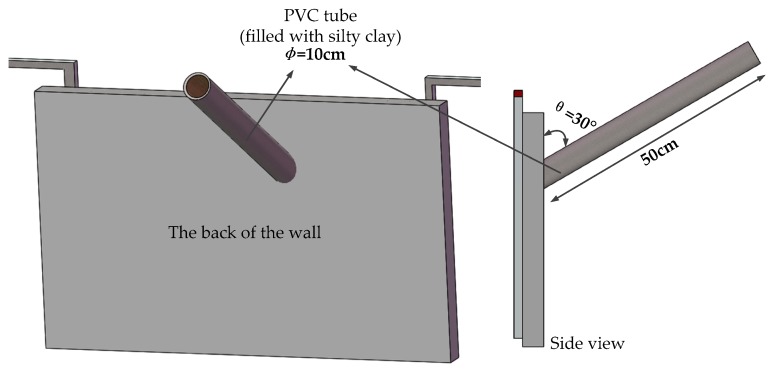
Drawing of the PVC connecting tube.

**Figure 7 sensors-19-02269-f007:**
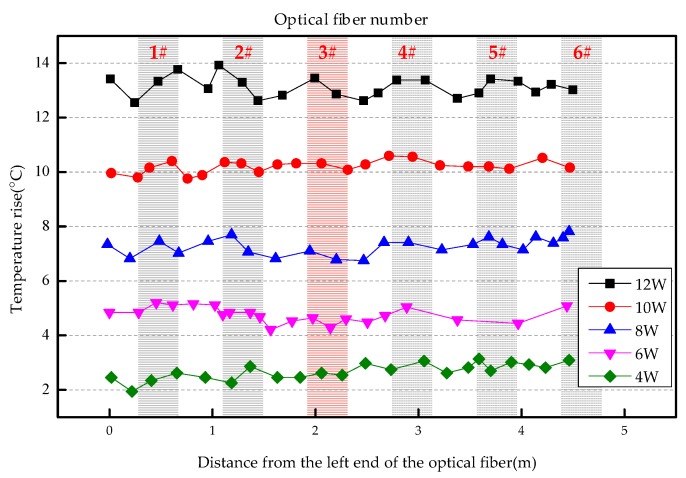
Temperature increase versus seepage flow velocity of 0 mm/h.

**Figure 8 sensors-19-02269-f008:**
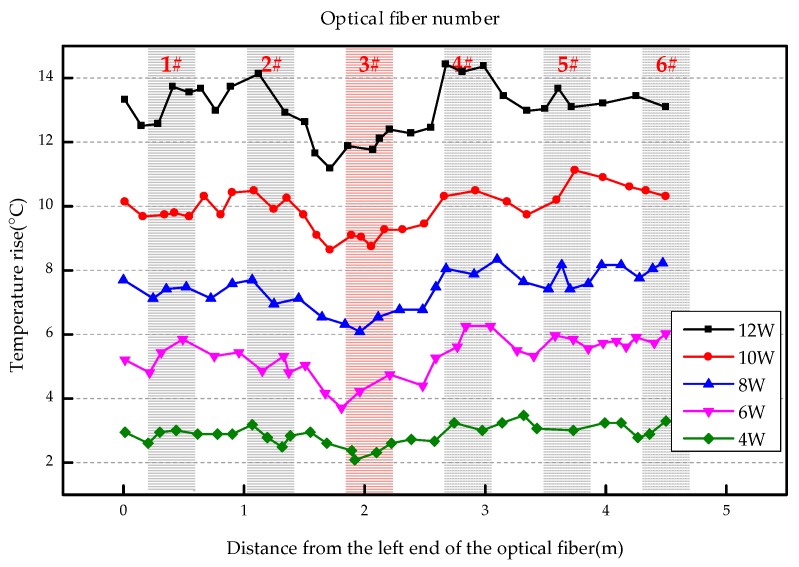
Temperature increase curve of seepage flow velocity at 50 mm/h.

**Figure 9 sensors-19-02269-f009:**
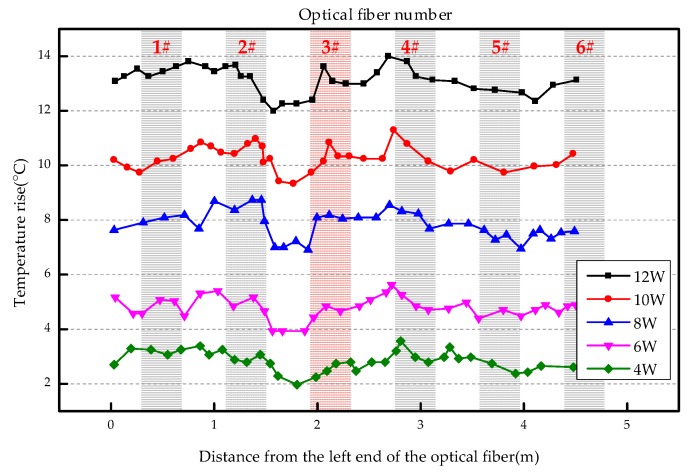
Temperature increase curve of seepage flow velocity at 100 mm/h.

**Figure 10 sensors-19-02269-f010:**
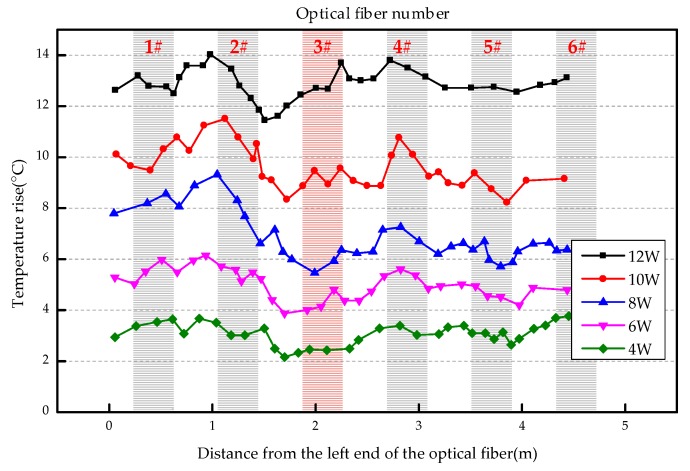
Temperature increase curve of seepage flow velocity at 150 mm/h.

**Figure 11 sensors-19-02269-f011:**
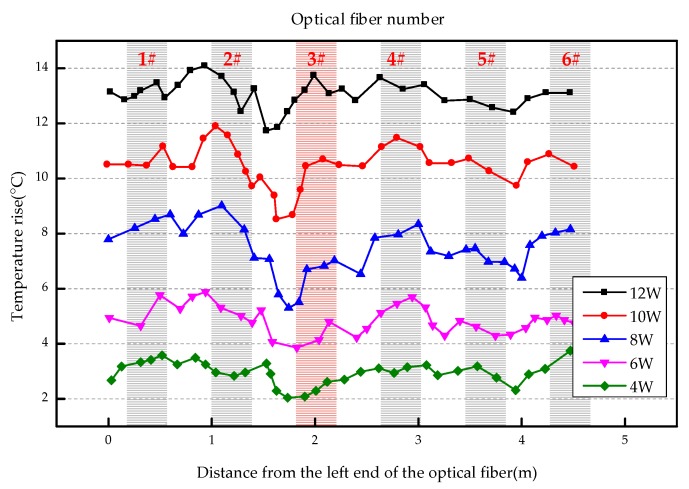
Temperature increase curve of seepage flow velocity at 200 mm/h.

**Figure 12 sensors-19-02269-f012:**
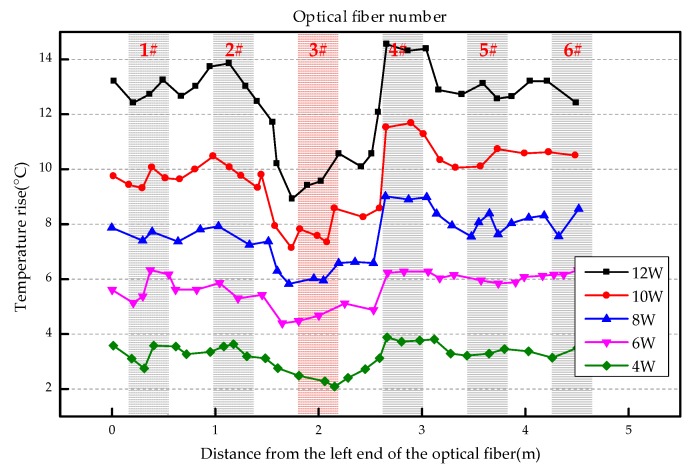
Temperature increase curve of seepage flow velocity at 250 mm/h.

**Figure 13 sensors-19-02269-f013:**
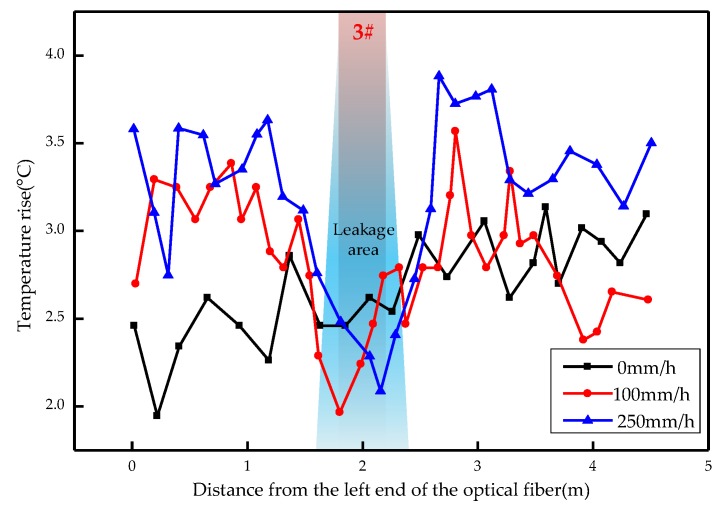
Temperature rise curve at different seepage flow velocity at 4 W.

**Figure 14 sensors-19-02269-f014:**
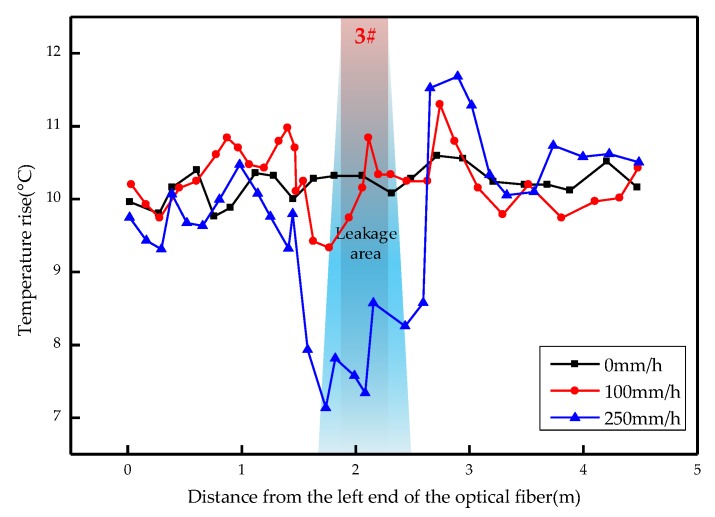
Temperature rise curve at different seepage flow velocity at 10 W.

**Figure 15 sensors-19-02269-f015:**
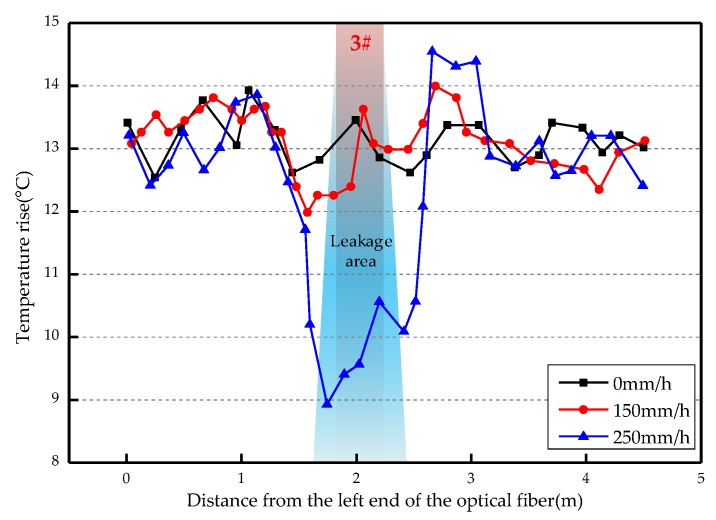
Temperature rise curve at different seepage flow velocity at 12 W.

**Figure 16 sensors-19-02269-f016:**
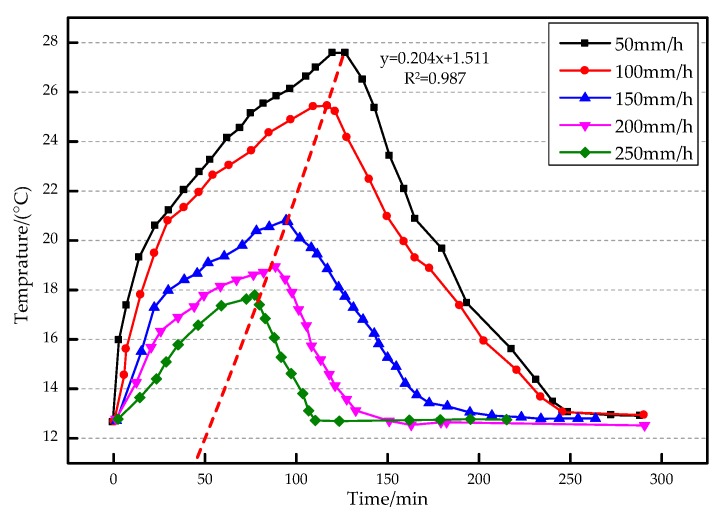
Relationship between temperature change and flow velocity of optical fibers at 12 W.

**Figure 17 sensors-19-02269-f017:**
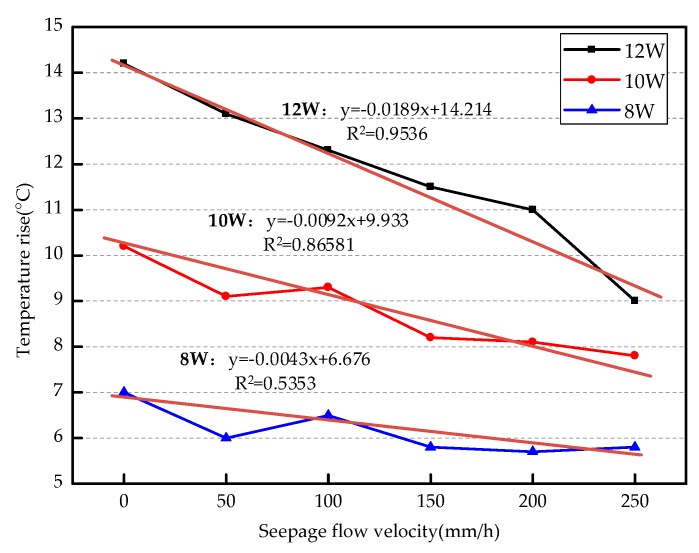
Relationship between temperature rise and seepage velocity.

**Table 1 sensors-19-02269-t001:** Parameters of used heating cable.

Material	Defined Temperature (°C)	Rated Power (W)	Rated Current (A)	Diameter (mm)	Line Resistivity (Ω/m)
Galvanized copper wire	65	10/15/25/30	40	8	67

**Table 2 sensors-19-02269-t002:** Basic parameters of used voltage regulator.

Material	Rated Power	Input Voltage	Output Voltage	Frequency	Insulation Grade
Full copper coil	200VA-40KVA	AC 220 V	AC 0–250 V	50–60 Hz	A

**Table 3 sensors-19-02269-t003:** Basic parameters of used DTS system.

Measuring Distance	Measuring Temperature Range	Accuracy	Temperature Resolution	Spatial Resolution
1-16km	−40–120 °C	±0.3 °C	0.1 °C	0.5–3 m

**Table 4 sensors-19-02269-t004:** Heating power and corresponding voltage.

4 W/m	6 W/m	8 W/m	10 W/m	12 W/m
90.48 V	110.82 V	127.96 V	143.07 V	156.72 V

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
