# Peer review of "Experimental Study of Leakage Monitoring of Diaphragm Walls Based on Distributed Optical Fiber Temperature Measurement Technology"

_sensors, 2019, doi:10.3390/s19102269_

Round 1

Reviewer 1 Report

This work deals with the testing of a distributed temperature monitoring system (DTS) for the leakage monitoring of diaphragm wall. The authors have employed a commercial DTS whose fiber was paired with a heating cable. The tests were conducted using a model and the relationship between fiber temperature and seepage speed was investigated.

In my opinion, the manuscript lacks scientific novelty as the sensing system is a commercial one. Moreover, the results seem still at preliminary stage as somehow declared by the authors themselves. Are the authors able to better highlight these two aspects, i.e. novelty of their solution and usefulness of their findings?

Few other comments/corrections:

1.       On page 2 (three lines before the page end) and Figure 1, Raman is wrongly typed.

2.       On page 3, I think that the superimposition of temperature field on fiber was done to INCREASE the SNR.

3.       Page 6, which is the DTS model?

4.       Page 7, line 6, the sentence “In this process…” is repeated twice.

5.       Page 10, line 6, “The” is repeated twice.

6.       Section 4.1, when commenting the Figures 7-12 and 13-15 please comment each one separately with explicit association between the comment and Figure number.

Author Response

    Thank you for your review. Your comments are very instructive.

     The scientific necessity of this research is that as far as I know, the distributed optical fiber system has not been applied to the seepage detection of diaphragm wall of foundation pit in soft soil area. It is more used in river embankments and other projects. The leakage of foundation pit is usually small, so it is difficult to accurately detect it with ordinary optical fiber temperature measurement system. This study attempts to explore the feasibility of using active heating optical fiber temperature measurement system in leakage detection of diaphragm wall. It provides a reliable theoretical basis for the application of distributed optical fiber temperature measurement system in the detection of diaphragm wall. We discussed this in more detail in the revised manuscript.

    Some responses to your other comments:

    The mistakes in comments (1) ,  (2) , (3), (4) and (5) have been corrected. Thank you for pointing out the mistakes for us.

     Figures 7-12 and 13-15 are separately explained.

     Thank you again for your guidance.

Reviewer 2 Report

Authors presented an optical fiber based leakage detection system using DTS. The basic principle is to monitor the temperature variation along the fiber and correlate it with a leakage. The paper is interesting with a good set of experiments. I recommend the publication of the paper pending some corrections:

- Authors should revise the paper and correct some errors on the English usage, such as grammar errors, repeated words and sentences as well as spelling errors, e.g., Roman line/Roman scattering.

- Please, define what is a C30 concrete block on Section 3.2. In addition, much more explanations needed for figures 4, 5 and 6. The authors barely mentioned these figures on the text. 

- Please, include on the caption of figures 7-12 that the leakage is concentrated on point #3.

- Authors should add a discussion about figure 9, where the temperature drop seems to be closer to point 2 than to point 3 (where the leakage occurred).

- I suggest the authors include a broader discussion about the sensor response time in Figure 16.

- Authors claim right after figure 17 that the curves presented in figure 17 are similar, which I disagree. If you look at the linear regressions presented, the slope of the curves are very different.

- What is the "fitting degree" that the authors mentioned on the last paragraph before the conclusions? Is it the correlation or determination coefficient?

Author Response

     Thank you for your review and comments, which are very valuable to us. Here are our response to your comments:

     (1) We have tried to correct the  grammatical and spelling errors in the paper Thank you for pointing out the errors .

     (2) In Section 3.2, we define C30 concrete, which is the strength parameter of concrete. We also increase the material composition and proportion of this kind of concrete.

     (3) Figures 4, 5 and 6 are explained separately.

     (4) In Figures 7-12,we explained that the leakage points are concentrated in No. 3 optical fiber. At the same time, we discuss the displacement of the leakage points in Figure 9, 10 and 11. The reason for this phenomenon is that the processing technology of the model box is not perfect, resulting in the uneven bottom of the model box. In the discussion of the results, the lowest temperature rise in the range of 0.4m on the left side of No. 3 is used as the actual leakage position.

     (5) We have considered using thermodynamic theory to discuss the heating up and cooling time in Figure 16, but because this is a dynamic process, it is impossible to calculate the heat taken away by the water flow and surrounding soil. Here we  discussed the seepage velocity and final stable temperature, heating power and test material parameters.

     (6) The slope of each curve in Figure17 is different. This is my misrepresentation. What we want to express is the linear fitting of each curve.

     (7) The fitting degree of the last paragraph before the conclusion represents the correlation coefficient of linear fitting, which is an inappropriate expression, and we have corrected it.

     Thank you again for your comments and guidance.

Round 2

Reviewer 1 Report

The authors have replied to the reviewers' comments in a satisfactory way. The revised paper is suitable for publication.

Reviewer 2 Report

Authors adressed all my suggestions. I recommend the publication of this work.